# Ecological Security Assessment Based on Ecological Footprint Approach in Hulunbeir Grassland, China

**DOI:** 10.3390/ijerph16234805

**Published:** 2019-11-29

**Authors:** Shanshan Guo, Yinghong Wang

**Affiliations:** School of Environment Science and Spatial Informatics, China University of Mining and Technology, Xuzhou 221116, China; guoshanshan@cumt.edu.cn

**Keywords:** ecological footprint, ecological capacity, ecological security, STIRPAT model, Hulunbeir grassland

## Abstract

Hulunbeir grassland, as a crucial ecological barrier and energy supply base in northwest China, suffers from a fragile ecological environment. Therefore, it is crucially important for Hulunbeir grassland to achieve the sustainable development of its social economies and ecological environments through the evaluation of its ecological security. This paper introduces the indexes of the ecological pressure index (EPI), ecological footprint diversity index (EFDI), and ecological coordination coefficient (ECC) based on the ecological footprint model. Furthermore, the Stochastic Impacts by Regression on Population, Affluence, and Technology (STIRPAT) model was applied to analyze the main driving factors of the change of the ecological footprint. The results showed that: The ecological footprint (EF) per capita of Hulunbeir grassland has nearly doubled in 11 years to 11.04 ha/cap in 2016, while the ecological capacity (EC) per capita was rather low and increased slowly, leading to a continuous increase of per capita ecological deficit (ED) (from 5.7113 ha/cap to 11.0937 ha/cap). Within this, the footprint of fossil energy land and grassland contributed the most to the total EF, and forestland and cropland played the major role in EC. The EPI increased from 0.82 in 2006 to 1.25 in 2016, leading the level of ecological security to increase from level 3 (moderately safe) to level 4 (moderately risky). The indexes of the EFDI and ECC both reached a minimum in 2014 and then began to rise, indicating that Hulunbeir steppe’s ecological environment, as well as its coordination with economy, was considered to be worse in 2014 but then gradually ameliorated. The STIRPAT model indicated that the main factors driving the EF increase were per capita GDP and the proportion of secondary industry, while the decrease of unit GDP energy consumption played an effective role in curbing the continuous growth of the EF. These findings not only have realistic significance in promoting the coordinated development between economy and natural resource utilization under the constraint of fragile environment, but also provide a scientific reference for similar energy-rich ecologically fragile regions.

## 1. Introduction

Ecological security means that a country or a region enriches ecological resources that could continuously meet social progress and economic development and lessens the restriction of the ecological environment on social and economic development [1]. However, with the acceleration of industrialization and urbanization, the increase of population, and a subsequent greater use of natural resources, ecological security issues, such as grassland degradation, forest decline, desert encroachment, and biodiversity loss are gradually becoming prominent. Maintaining a dynamic equilibrium between humans and nature becomes particularly hard and important in ecologically fragile areas [2]. This requires to research the relationship between human activity and environmental change, optimize the structure of resource utilization, and minimize the impact of human behavior on the ecological environment under the premise of maintaining sustainable economic growth and safeguarding human well-being [3,4].

To further understand the change of the ecological environment, many beneficial explorations in terms of concept connotation, evaluation framework, and control approach of ecological security have been conducted [5,6,7]. The related researches have ranged from the empirical study of environmental change and security to the interactions between them, emphasizing the harmonious coexistence of socio-economic development and ecological security [1,8]. Additionally, various approaches have been applied to evaluate the state of ecological security, such as the pressure-state-response [9], landscape ecology [10,11], material/substance flow analysis [12,13], ecosystem services [14,15], and ecological footprint [8,16,17]. Among which, the ecological footprint method is the most widely used. It is defined as a bio-productive land area that maintains human living needs while absorbing pollution caused by human activities. The primary advantages of this method are that it is easy to apply, repeatable, and simple to understand [18,19]. Scholars have used this method to study multiple scales of ecological footprints. On a large scale, Wackernagel et al. [20] researched the ecological footprints of 52 countries that include 85% of the global population and 95% of the global economic output, finding that 35 of these countries were experiencing ecological risk. Niccolucci et al. [21] studied the ecological footprints of 150 countries from 1961 to 2007 by dividing the change tendencies of ecological footprint and ecological capacity into four types: parallel, scissors differential, wedge, and downtrend. Li et al. [19] researched the five countries in arid areas of Central Asia, showing that the ecological pressure in all five countries was rising, and the overall ecological security situation changed from comparably safe before 2000 to unsafe in 2005 and then may be at risk in 2025.

However, these large-scale studies generally ignored the differences in resources, technologies, and other aspects among the various regions. In response to this gap, some scholars have recently researched the ecological footprint based on comparatively smaller scales of province, city, and county. For example, Dong et al. [22] evaluated the ecological security and natural capital utilization of Hainan Province, China, from 2005 to 2016, uncovering the main factors influencing the changes of ecological footprint by partial least squares regression model. Pan et al. [23] analyzed the dynamic changes in supply and demand of resources and revealed that China’s Shanghai is suffering a high ecological footprint intensity and a poor coordination relationship between its economy and environment. Zhao et al. [24] evaluated the ecological security of Lhaze County in China’s Tibetan Autonomous Region, showing that ecological environment has deteriorated from “early stages of damage status” in the 1980s to “moderately damaged status” today.

Scholars have also used different models to analysis the main factors that affect the change of the ecological footprint, such as the Stochastic Impacts by Regression on Population, Affluence, and Technology (STIRPAT) [25], partial least squares regression (PLS) [26], and log-mean divisia index (LMDI) [27]. One of the most representative methods for exploring the influencing factors is the STIRPAT model. Its primary advantage is that it allows for non-monotonic or non-proportional effects from the driving forces [28]. Currently, this model has been widely used in studies to analyze the effect of population, affluence, and technology on environment.

Affected by climate change and human activities, mining areas often overlap with fragile ecological regions [29]. Hulunbeir grassland, as an important natural ecological barrier and energy supply base in north China, provides a significant contribution to the country’s economic development and ecological security. However, global climate change and continuous socio-economic development with excessive exploitation of resources have jointly induced soil and water loss, grassland desertification, degradation, and salinization [13]. Eco-environmental issues have been paid close attention by the whole society [30]. At present, the pertinent literatures in this area mainly focuses on the discussion of the variation tendency of grassland production capacities [31] and correspondingly ecological protection measures [32]. While research rarely shows how natural resources are used and ecological security changes over time, these exhibit the remarkable effects of ecological policy implementation [33]. Especially since the eleventh “Five Plan”, Hulunbeir city has further strengthened ecological restoration and made efforts towards industrial transformation. Therefore, in this paper, the ecological footprint (EF) framework and the ecological capacity concept are introduced to clarify the supply and demand of the regional ecosystem. Three indicators of ecological pressure index (EPI), ecological coordination coefficient (ECC), and ecological footprint diversity index (EFDI) from 2005 to 2016 are calculated to provide a more comprehensive assessment of historical and current ecological security of mining areas in Hulunbeir grassland. Additionally, the STIRPAT model is constructed to further study the main socio-economic driving factors that lead to the emergence or even aggravation of ecological deficits and quantify the importance of each driving factor. This study can provide a scientific basis for solving the contradiction between the rapid development of the social economy and degradation of ecological environment and thereby provide a reference for ecological environment management decisions for mining areas in arid and semi-arid grasslands in China. 2. Study Area and Data Sources.

### 1.1. Study Area

Hulunbeir grassland is mostly distributed in Hulunbeir city (E 115°31′–126°04′, N 47°05′–53°20′), northeast of Inner Mongolia (Figure 1). Except for the transition area of forest and grassland in the eastern region, the rest are basically natural grasslands with an area of approximately 11.27 × 10^4^ km^2^, covering about 11.54% of Inner Mongolia’s total grassland area. Hulunbeir grassland as a whole is a plateau landform with an altitude of 650–1000 m. It belongs to a continental arid to semi-arid climate with an annual average temperature of −3–0 °C and annual precipitation of approximately 350 mm. Hailaer River, Yimin River, and Gen River originating from the Greater Khingan Mountain are the major water supplies for grasslands. From the western foothills of the Greater Khingan Mountains to the Mongolian Plateau, there is a distribution of zonal grasslands with arid steppe, meadow steppe, and forest steppe. The perennial herbaceous community is the basic feature of the grassland ecology in this region, with about 1000 plant species. Moreover, Hulunbeir grassland is rich in coal resources, with an area of about 2.7 × 10^4^ km^2^ accounting for 31% of the total region. Since the reform and opening up of China, the coal industry has undergone a rapid development. Especially in 2006, the coal mines of Jalainur, Baorixile, and Yimin were listed in the second batch of 26 national planned coal mining areas [34], with the coal industry expanding vigorously. At present, its proved reserves of coal resources reach 29.785 billion tons, which is 1.8 times higher than the combined storage of the three provinces in Northeast China (Heilongjiang Province, Jilin Province, and Liaoning Province) [35], and more than 370 mining sites and 355 mining enterprises of various categories have been constructed. The accumulated solid waste output produced by mining is about 370 million tons, and the total area of mined-out subsidence area is approximately 42 km^2^ [36]. All of these problems are concentrated in the grassland area, such as Hailar Basin, Labudalin, and Miandu River. The area of the degradation, desertification, and salinization in Hulunbeir grassland has increased from 13% in the 1960s to 21% in the 1980s, to about 30% in the 1990s and to nearly 50% in the beginning of this century [37], which seriously handicaps the sustainable development of the regional economy and environment.

### 1.2. Sources of Data

Hulunbuir’s rapid economic growth is mainly supported by a high consumption of energy resources. There are three large mining areas listed in the second batch of the 26 nationally planned coal mining areas in 2006, which have promoted the rapid development of energy-related industries. At the same time, the new eleventh “Five Plan” emphasizes the importance of ecological environment protection and the necessary of industrial transformation to reduce dependence on energy-intensive industries. The disparity between economic development and environmental protection is becoming increasingly acute. Based on this situation, we mainly select the time series of 2006–2016 to explore the dynamic change of the ecosystem’s health and natural resource utilization after the new eleventh “Five Plan”. Detailed data sources and descriptions are shown in Table 1.

## 2. Methods

### 2.1. Construction of Ecological Security Evaluation Framework

Here, we selected four indicators of Flux (F), Pressure (P), Diversity (D), and Coordination (C) on the basis of the method of ecological safety evaluation (Figure 2). F reflects the difference between ecological footprint capacity (EC) and ecological footprint (EF). It indicates changes on an ecosystem’s support to socio-economic system, thereby reflecting ESS from an absolute value change perspective. F is the equivalent of the ecological balance, which can be calculated through the classic EF model. The ecological pressure index (EPI), as a substitute for P, is the pressure level on the ecosystems generated by a given socio-economic system or a specific population scale. It is defined as the ratio of per capita EF (*ef*) to per capita EC (*ec*). EPI reflects ESS by analyzing stress suffered by ecosystem per unit of ecological capacity. Diversity (D) expresses both the amount and the distribution of the different EF components, which reflects ESS from an EF composition and structure perspective. Finally, Coordination (C) is used to measure the coordination degree between socio-economic development and ecosystems. The ecological footprint is affected by population, economy, and technology. Thus, we created multiple regression equations between the ecological footprints and variables to explore the main driving factors of the ecological footprint change.

### 2.2. Evaluation Model of Ecological Footprint

The ecological footprint is the area of productive land and water needed to support the regional population and the land needed to absorb the waste produced by those populations [41]. This approach provides a kind of simple methodology but comprehensive way to measure direct and indirect human consumption on the regional regenerative capacity. Then, by comparing it with the biocapacity available, we can judge whether the development pattern of this region is in a sustainable state. According to the theory of ecological footprint, the biologically productive land can be divided into 6 types, i.e., cropland, grassland, forestland, water, fossil energy land, and build-up land [20], each type has a different production capacity of per unit area. The specific calculations formulas are as follows:
(1)$$EF=N\times ef=N\times \overset{n}{\underset{i=1}{\sum }}\left(aa_{i}\times r_{i}\right)=N\times \overset{n}{\underset{i=1}{\sum }}\left(\frac{c_{i}}{p_{i}}\times r_{i}\right).$$

In the formula above: EF is the total ecological footprint (ha); N is the total population; i is the category of items consumed by a certain population (i = 1,2,…, *n*); ${aa}_{i}$ is the ecologically productive area from the ith consumption item; $p_{i}$ is the average productivity of the ith item in a certain area (kg/ha); $c_{i}$ is the per capita quantity of the ith item (kg/ha) affected by the productivity and trade balance amount; $r_{i}$ is an equivalence factor, which describes the ratio of the productive capacity of a certain type of bioproductive land to the productive capacity of all the world’s bioproductive land. The equivalence factor of each biologically productive land is shown in Table 1

(2)$$EC=N\times ec=N\times \left(1-12\%\right)\sum_{j=1}^{6}\left(a_{j}\times r_{j}\times y_{j}\right)$$

In the formula above: EC is the total ecological capacity (ha); j represents the area of the biologically productive land required; $a_{j}$ represents the per capita area of the biologically productive land for items of the *j*th category (ha); $y_{j}$ is yield factor, which describes the ratio of average land productivity of a country or region to the global average productivity of the same land type. The yield factor of each biologically productive land is shown in Table 1. In addition, the area of biologically productive land should be decreased by 12% to account for biodiversity conservation [42].

### 2.3. Evaluation Model of Ecological Security

#### 2.3.1. Ecological Deficit/Surplus

As the EF and EC are both measured by the area of biologically productive land, they can be compared directly. Ecological deficit/surplus (ED/ES) presents the profit and loss of supply and the demand situation of the regional ecological system [17]. When a region’s EC is less than the EF in a region, an ecological deficit (ED) appears, indicating that the supply of regional ecological resources neither meet the demands of social development nor bear the corresponding environmental purification and renewal. Therefore, the region may import resources from surrounding cities or even other faraway cities to satisfy increasing local demand for natural resources and energy. Conversely, ecological surplus (ES) indicates that the supply of regional ecological resources is sufficient to meet the needs of human production. The formula for the ED/ ES is as follows:(3)ED/ES=EF−EC=N×(ef−ec)

#### 2.3.2. Ecological Pressure Index

The ecological pressure index (EPI) mainly reflects ecological pressure which is caused by the consumption of resources and sequestering carbon dioxide emissions, and so forth, in the industry and daily life of local residents [8], representing the pressure intensity suffered by regional ecological environment. If 0 < EPI < 1, EPI will be positive and the supply of ecological resources exceeds the demand for it, indicating that the ecological security remains in a sustainable status. If EPI = 1, the supply of the ecological resource and the demand for it are equal, indicating that the ecological security is in a critical status. Finally, if EPI > 1, people’s demand for ecological resources is greater than its supply, indicating that the regional ecology is in a threatened status. Using data calculated for the EFs of 147 countries or regions, as provided in the Living Planet Report by the International Monetary Fund in 2004, Yuan [43] and Chu [17] detailed the classification standard of ecological security (Table 2). The formula is shown as follows:(4)EPI=ef/ec.

#### 2.3.3. Ecological Coordination Coefficient and Ecological Footprint Diversity Index

Ecological deficit is an absolute value and cannot reflect its relationship with resource endowment conditions. Therefore, it is necessary to introduce the concept of an ecological coordination coefficient (ECC) to compensate for this deficiency in the ecological deficit [4]. Ecological coordination coefficient represents the coordination degree between regional ecological environment and socio-economic development. The formula is as follows:(5)ECC=ef+ecef2+ec2=efec+1efec2+1=EPI+1EPI2+1.

Due to *ef* and *ec* being larger than 0, the ECC ranges from 1 to 1.414. The closer it is to 1.414, the better the coordination. Conversely, the closer it is to 1, the worse the coordination.

The ecological footprint diversity index (EFDI) reflects the abundance of different land types and the fairness of ecological footprint distribution in a region [16,44]. The more equal the ecological footprint distribution in an eco-economic system, the higher the ecological diversity is for the ecological economy of given system components. Generally speaking, in the early stage of regional development, the EFDI is relatively low. However, with the development of social economy, the diversity index gradually increases, which can promote the improvement of energy utilization efficiency. The formula is as follows:(6)$$EFDI=-\sum p_{i}\ln p_{i}.$$

In the formula above: $p_{i}$ is the proportion of the *i*th category of land type in regional ecological footprint.

### 2.4. The STIRPAT Modelling Approach

Ehrich and Holdren [45] firstly proposed the IPAT (Environmental Impact, Population, Affluence, Technology) model to analyze the relationship between population, economy, technology, and environment. While the weakness of the IPAT model is that it regards population, economy, technology, and environmental issues as change relations in equal proportion, this is inconsistent with the reality. In addition, the importance degree of each driver cannot be clearly judged. To overcome these problems, Dietz and Rosa [46] proposed the Stochastic Impacts by Regression on Population, Affluence, and Technology (STIRPAT) model, making quantitative analysis of environmental problems more flexible. The generic STIRPAT model is given as:(7)$$I=aP^{b}A^{c}T^{d}\xi \;.$$

In the formula above: I represents the environmental impact; a and ξ are the coefficient and random error of the model; P, A, T represent population, affluence, and technology, respectively; b, c, and d denote the exponentials of the driving forces.

In order to measure hypotheses and assess the importance of each influencing factor, the STIRPAT model is taken as a logarithm to get a linear model [25]:(8)lnI=lna+blnP+clnA+dlnT+lnξ

However, the accuracy of the evaluation results will be reduced because of the multicollinearity among socio-economic variables [47]. Therefore, in this study, the principal component analysis method (PCA) was used to improve the STIRPAT model. The basic principle is: (1) Standardizing the original data and then calculating the correlation coefficient matrix; (2) obtaining the eigenvalues and variance contribution rates of the correlation matrix, and determining the principal components; (3) sing principal components to replace the original variables for multiple regression *F*; (4) substituting the original variables into the principal component regression model; (5) based on the results of the principal component analysis, the logarithms of the extracted principal components were taken respectively and then restored to the original variable form in the STIRPAT model.

## 3. Results and Discussion

### 3.1. Evaluation Results of Ecological Footprint and Ecological Capacity

The *ef* of the study area increased from 5.71 ha/cap in 2006 to 11.04 ha/cap in 2016, at a range of 93.34% (Table 3). From 2006 to 2014, the average annual rate was 8.01%, while, from 2015 to 2016, the value of the *ef* showed a negative growth. This could be due to the effect of the global coal market and emerging energy, as well as the regulation of illegal and irregular mining enterprises (37 mine enterprises were closed in 2015) in Hulunbeir grassland. In terms of components of the *ef*, the change in the water *ef* and build-up land *ef* over time was not obvious, but the the *ef* of fossil energy land and cropland increased obviously, especially in 2010 to 2014, the fossil energy *ef* rose from 2.94 ha/cap to 5.29 ha/cap, which could be resulted from the boom of industrial enterprises above designated size (increased nearly 100 over those 4 years). Similarly, the *ef* of grassland and cropland increased by 1.75 ha/cap and 0.65 ha/cap respectively over these 11 years, which reflected the increase of people’s consumption of milk, meat, and agricultural products and that the living standard was gradually improving. In addition, the *ef* of fossil energy land exceeded the *ef* of grassland since 2007 and became one of the most important issues in daily life in recent years. In other words, the EF in the study area has increasingly come from energy-related production’s consumption from then on. On the contrary, the forest *ef* showed a declining trend in recent years, mainly because the study area implemented effective measures against the commercial logging of natural forest since 2012 and cancelled the original production index of 320,000 m³ of natural forest commodity timber, which significantly reduced the consumption of wood.

Table 4 shows that the per capita EC of the study area experienced fluctuant growth from 6.81 ha/cap to 8.85 ha/cap throughout the study period, with an average annual rate of only 1.03%. But compared with the overall decreasing trend in China [16], the slight increase of the *ec* in Hulunbeir grassland indicates that the implementation of ecological construction projects have played a positive role in improving the supply capacity of regional resources. Specifically, as the biological productivity of cropland and forestland was higher than that of other land use types, the *ec* of these two land use types accounted for the largest proportion (58% and 33.55%, respectively) in Hulunbeir grassland, followed by build-up land (4.64%), grassland (3.39%), and water area (0.42%). Additionally, the *ec* of cropland grew fastest, from 2.15 in 2010 to 3.83 in 2016, indicating that the reclamation intensity of cropland in the study area was strengthened and the other land utilization types (mainly grassland) had been converted into cropland. The *ec* of forestland and grassland both experienced a trend of decreasing first and then increasing, mainly because of a series of treatment measures, such as managing water pollution, enclosure, returning grazing land to grassland, and the prohibition of commercial logging.

### 3.2. Evaluation Results of Ecological Security

Based on the results of the *ef* and *ec* in study area, we obtained per capita ED/ES of the six productive land types, shown in Figure 3. The *ec* was larger than the *ef* during 2006–2009, meaning that during these 4 years, the ability of regional natural resources that supported human activities still remained in a sustainable state as a whole. However, the *ef* exceeded the *ec* in 2010, and then the per capita ES gradually transformed into per capita ED with the increase of the fossil energy consumption and the grassland deterioration. This indicated that the equilibrium between demand and supply showed an unbalanced status and the unsustainable tendency was increasingly obvious in Hulunbeir grassland. Furthermore, the cropland, forestland and build-up land were in a state of ecological surplus during the past eleven years, and the bearing capacity of forest land experienced a small upward trend after the implementation of various protection policies. Build-up land and water area distributed near the boundary between deficit and surplus changed slightly. However, the fossil energy land and grassland showed an obvious deficit, and their change trend was basically consistent with the per capita ED, indicating that these two components contributed the most to the change of per capita ED.

Trends of ecological pressure index (EPI), ecological footprint diversity index (EFDI) and ecological coordination coefficient (ECC) in Hulunbeir grassland from 2006 to 2016 were shown in Figure 4. As a whole, the reverse trend of ECC and EPI was obvious. The coordination of ecological environment was good when ecological pressure was low; conversely, it was poor when ecological pressure was high. Therefore, the threshold value of EPI could be determined by the critical value of ECC, and then the ecological security level and ecological security alarm level could be determined. Specifically, EPI rose from 0.83 in 2006 to 1.36 in 2014 after which it showed a slight downward trend from 1.36 in 2014 to 1.25 in 2016. In 2010, it was greater than 1, which meant that the degree of security increased from level 3 (moderately safe) to level 4 (moderately risky) (Table 5). However, from 2014 to 2016, although ecological security was still at level 4, the EPI decreased obviously, indicating that the ecological security condition in the study area showed an evident improvement. It could be related to the “Three Horizontal and Three Vertical” industrial transformation and development strategy implemented by governments, in 2014, and grassland ecological protection measures, such as returning grazing to grassland and enclosure projects.

In addition, with an increase in the ecological deficit and ecological pressure, the EPI reached its maximum in 2014, indicating that the degree of the use of natural resources, and the rate at which waste was being released, had already exceeded the system’s recycling and self-purification capabilities. The ECC declined slightly over the study period, indicating the coordination between regional ecological environment and socio-economic development was gradually decreasing. In 2014, it reached the minimum, contrary to the EPI, indicating that when the ecological pressure rises to a certain point, the ecological environment will become uncoordinated and unsafe.

The EFDI showed a downward trend from 2006 in 1.42 to 1.25 in 2016. At the same time, we can see that the annual average decline rate of EFDI during the period 2006–2013 was 1.4%, the increase rate of EPI was 5.6% at the same time. However, when the annual average decline rate of EFDI gradually approached 0, the change rate of the EPI began to decline obviously. What is more, when the EFDI reached the minimum in 2014, the EPI reached the maximum. Therefore, an increase of ecosystem diversity by using different types of land resources equally and improving resource utilization efficiency could help alleviate ecological stress and enhance the development capability of the ecological system.

### 3.3. Identification of Driving Factors of Ecological Footprint

Climate, precipitation, soil, and other factors have to go through a long evolution process before they can affect the ecological footprint [22,48], which could be ignored for the time being. Therefore, what we explore are internal drivers of the ecological footprint primarily skewing towards the impact of regional economic and social development. On the basis of the STIRPAT model, the index of energy footprint driving forces in study area was selected from three aspects of economy, society, and technology. Economic growth requires a large amount of resources and energy from nature. Meanwhile, it will produce various emissions of pollution, leaving a deep imprint on the natural ecology and further affect the change of ecological footprint. Therefore, we selected per capita GDP (A) (represents the comprehensive situation of regional economic development) and the proportion of secondary industry (T) (highly relies on energy and raw materials and generates more wastes) to represent the ecological footprint consumption of economic growth. Social development is another important aspect that determines the change of ecological footprint, mainly reflected by population scale and consumption structure [22,49]. Therefore, the year-end resident population (P) and urbanization rate (U) were selected to explore the effect on the ecological footprint. Generally, the improvement of science and technology levels will reduce the energy consumption, that is, the unit of energy consumption will a produce greater value, and the waste rate of resources will be reduced. Hence, we adopted the unit GDP energy consumption (C) to reflect the technology development level.

Initially, the correlation matrix method was utilized to check the correlation coefficients among variables (Table 6). It is obvious that the collinearity among these independent variables was higher because many correlation coefficients were greater than 0.8, indicating that there was serious multicollinearity existing among them. This would impact the accuracy and credibility of STIRPAT model. Thus, it was necessary to adopt the principal component analysis (PCA) to eliminate this effect.

In order to avoid the influence between variables, Kaiser–Meyer–Olkin (KMO) test of sample data should be conducted before principal component analysis (PCA). Results showed that the KMO measurement value was 0.695, greater than 0.5, the Bartlett sphericity test value was 71.849, and the sig < 0.001, passing the test, which indicated that PCA was feasible. According to the results of PCA (Table 7), when the first two principal components were extracted, the cumulative contribution rate was 97.332%, greater than 85%, indicating that it contained 97.332% of the original variable information and could replace the original variable to achieve a satisfactory effect.

As showed in Table 8, the principal component F1 was mainly related to the total population, urbanization rate, and unit GDP energy consumption, and the variance percentage was 48.908%. The principal component F2 was mainly related to the per capita GDP, the proportion of the secondary industry and unit GDP energy consumption, and the variance percentage reached 48.425%. On the basis of the results of the principal component analysis, the two extracted principal components, F1 and F2, were respectively expressed by the influence factors after the logarithm, and then restored to the original variable form in the STIRPAT model, shown as follows:(9)$$F_{1}=-0.135LnA-0.384LnT+0.584LnP+0.585LnU-0.121LnC$$
(10)$$F_{2}=0.459LnA+0.675LnT-0.286LnP-0.284LnU-0.206LnC$$
(11)$$LnI=0.795\;+\;1.160F_{2}+LnK.$$

The original time series data of the ecological footprint were converted into a natural logarithm, and the converted data were represented by *LnI*. With the dependent variable *LnI* as the control variable and the comprehensive variables F1 and F2 as the explanatory variables, ordinary least square regression was adopted to conduct regression analysis on the variables based on SPSS19.0 software. The regression results and equation test are shown in the Table 9. The R^2^ and adjusted R^2^ were both greater than 0.9, and the F statistic value is significant at the level of less than 0.01, indicating the overall fit was very good. According to the regression coefficients, the regression equations of dependent variable *LnI* and comprehensive variables F1 and F2 were obtained, shown in Equation (11).

After the reduction of Equation (11), the expression of Hulunbeir grassland’s ecological footprint and five driving factors based on STIRPAT model was obtained:(12)$$I=NA^{0.425}T^{0.478}P^{0.135}U^{0.136}C^{-0.335}$$

Results of STIRPAT model indicated that the per capita GDP, the proportion of the output value of the secondary industry, the total population, as well as the urbanization rate in the study area are positively correlated with the ecological footprint, with force indexes 0.425, 0.478, 0.135, and 0.136 respectively. Among them, the proportion of the secondary industry had the largest effect on ecological footprint. Every 1% increase in the proportion of secondary industry would cause an increase of 0.478% in the total EF. With the rising of per capita GDP, the proportion of the secondary industry experienced a rapid increase, from 27.6% in 2006 to 44.7% in 2016. The development of the secondary industry is often accompanied by the consumption of energy resources and the discharge of wastes, putting great pressure on the local environment. The rapid growth of the secondary industry will undoubtedly lead to the increase of the regional ecological footprint. Therefore, the optimization of industrial structure is an important factor to improve the quality of the ecological environment.

In addition, the positive driving effect of the population and urbanization rate on the ecological footprint was small. Every 1% increase in the population and urbanization rate would cause an increase of 0.135% and 0.136% in the EF, respectively. According to the statistics, the urbanization rate increased slowly during 11a, from 65.72% in 2006 to 71.52% in 2016; the year-end resident population dropped from 2.7 million in 2006 to 3.4 million in 2016. This was consistent with the analysis results of the STIRPAT model, indicating that the population and city size of Hulunbeir grassland had little impact on the change of the EF. The results of this study are inconsistent with Zheng’s research [50], i.e., urbanization rate and ecological capacity are the main factors affecting the ecological footprint in China, but they are similar to Yang’s results [51], i.e., population and urbanization rate have little influence on the ecological footprint, but the output value of the secondary industry has a great influence in energy-rich ecologically fragile regions.

On the contrary, the technology level (unit GDP energy consumption) was the main negative driving force on the growth of the EF, and every 1% increase in technology level may induce about 0.335% decrease in the EF, supplementing Yang’s [52] theoretical analysis on the inhibitory effect of technical factors on the growth of the ecological footprint to a certain extent, i.e., the improvement of science and technology provides clean technologies and production processes for industrial production, which could, theoretically, reduce resource consumption, decrease the amount of contaminant emission, and promote the utilization of renewable energy resources.

## 4. Conclusions

The EF framework used in this study offers an intuitive to rationally judge the relationship between regional socio-economic development and ecological capacity from a supply-and-demand perspective. Based on this theory, the indexes of the EPI, ECC, and EFDI were obtained to determine the level of ecological security and coordination between the ecosystem and economy. The main conclusions are as follows:

The per capita EF of Hulunbeir grassland nearly doubled during the past 2006–2016 years. While, the per capita EC was rather low and increased by only 29.9%, the footprint of fossil energy land contributed the most to the total EF, followed by grassland, cropland, forestland, build-up land, and water. In terms of time variation characteristics, the EF exceeded the EC in 2010, and the ecological deficit began to show and gradually expand from then on. At the same time, the EPI increased obviously, causing the degree of ecological security to rise from level 3 (moderately safe) to level 4 (moderately risky) and the alarm level changed from low alarm to moderate alarm. Additionally, the value of the EFDI and ECC dropped to different extents during 11a, but both reached a minimum in 2014 and have increased slightly since. Contrarily, the EPI reached a maximum in 2014 and then decreased slightly. These changes show that the status of ecological environment and its coordination with economy were in a worse position in 2014 but have been gradually alleviated since.

Driving force analysis shows that the per capita GDP and the proportion of secondary industry are mainly positive driving factors of EF growth. However, technological advances played an important role in curbing the growth of the EF during the study period. The increase of the population and city size had little influence on the ecological footprint. Therefore, Hulunbeir city should continue to strengthen the optimization and upgrading of the industrial structure, enhance the capacity of technological innovation and promote the use of clean energy. At the same time, ecological environment restoration should be strengthened to improve the carrying capacity of the environment.

Although the ecological footprint model has been widely used in ecological environment and sustainable development research, the parameters involved in the model, such as the equivalence factor, yield factor, and average yield, are various and the comparison scale is different (global scale, national scale, and regional scale), which may have a certain impact on the evaluation results. Therefore, it is necessary to further optimize the model to achieve a more comprehensive regional ecological assessment in future studies.

## Figures and Tables

**Figure 1 ijerph-16-04805-f001:**
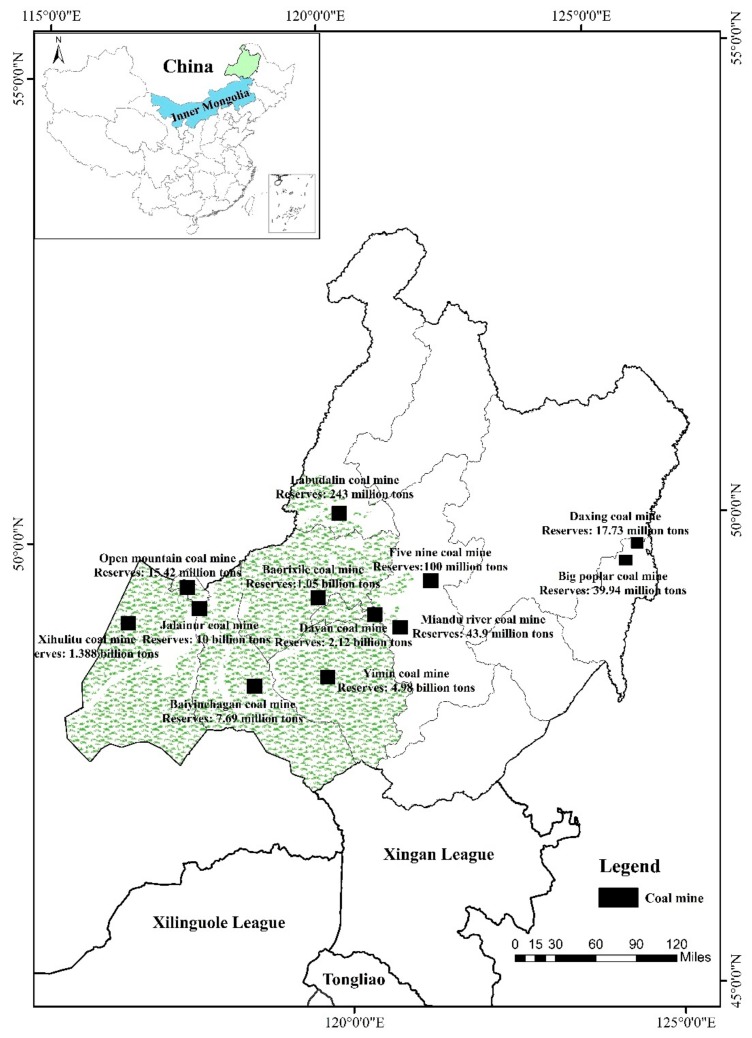
Location of the overlapped areas of grassland and coal resources.

**Figure 2 ijerph-16-04805-f002:**
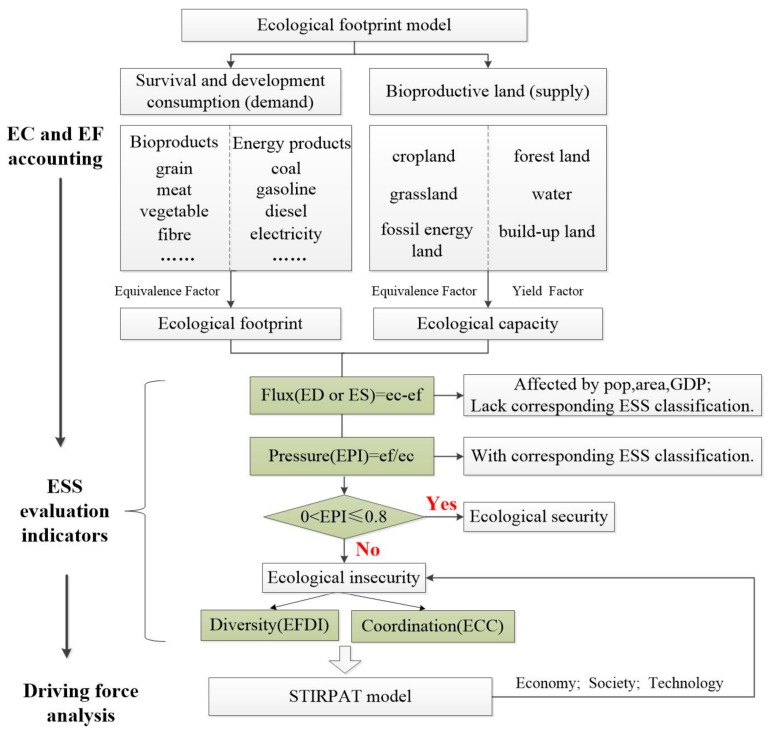
The framework of ecological security evaluation.

**Figure 3 ijerph-16-04805-f003:**
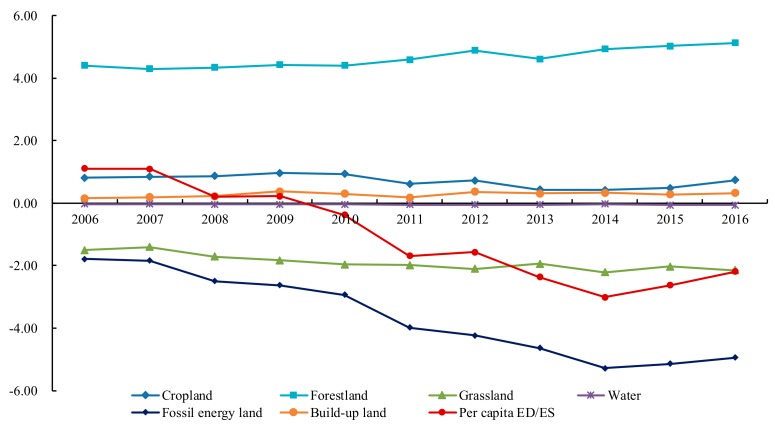
Changes of per capita ecological deficit/ecological surplus in Hulunbeir grassland from 2006 to 2016.

**Figure 4 ijerph-16-04805-f004:**
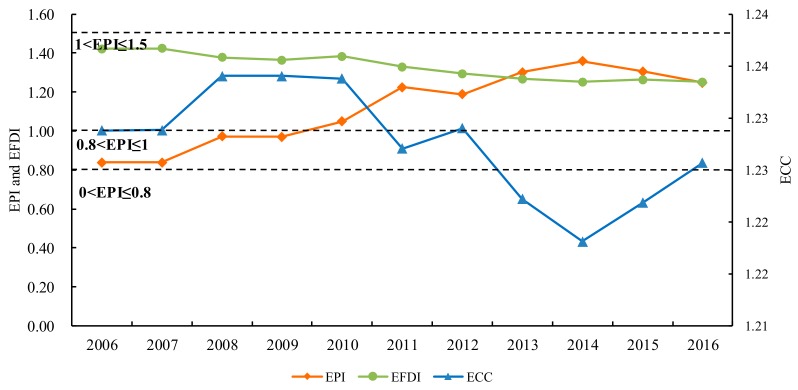
Evolution of ecological pressure index, ecological coordination coefficient and ecological footprint diversity index in Hulunbeir grassland from 2006 to 2016.

**Table 1 ijerph-16-04805-t001:** Indicators and data sources.

Items	Indicators	Data Sources
Biological account	Agricultural products: wheat, corn, rice, sorghum, potato, oil crop, vegetables, beans, wine, sugar, pork and eggs Forest products: fruits and wood Grass products: beef, lamb, poultry, milk, dairy products, sheep wool, goat wool, cashmere Aquatic products: freshwater	«Hulunbeir Statistical Yearbook» (2007–2017)
Energy account	The consumption of raw coal, crude oil, coke, gasoline, kerosene, diesel oil, fuel oil, electricity, heat	«Hulunbeir Statistical Yearbook» (2007–2017)
Land use	Land use area	Land Resources Data of the Ministry of Natural Resources (2006–2016) and «Hulunbeir Statistical Yearbook» (2007–2017)
Equivalence factor	cropland (2.8), grassland (0.5), forest land (1.1), water (0.2), fossil energy land (1.1), build-up land (2.8)	«Calculation of China’s equivalence factor under ecological footprint mode based on net primary production» [38]
Yield factor	cropland (1.7), grassland (0.19), forestland (0.91), water (1), fossil energy land (0), build-up land (1.7)	«Calculating national and global ecological footprint time series: resolving conceptual challenges» [39] «Quantitative analysis of sustainability development of inner Mongolia» [40]
Population, economy, and technology	Population: year-end resident population, urbanization rate Economy: per capita GDP, proportion of secondary industry output value Technology: unit GDP energy consumption	«Hulunbeir Statistical Yearbook» (2007–2017) «Inner Mongolia statistical yearbook» (2007–2017) National economy and society developed statistical bulletin in Hulunbeir (2006–2016)

**Table 2 ijerph-16-04805-t002:** Classification standard of ecological security.

Ecological Security Grade	Range of EPI	Characterization State	Ecological Security Alarm Level
1	<0.5	Pretty safe	No alarm
2	0.50–0.80	Safe	
3	0.81–1.00	Moderately safe	Low alarm
4	1.01–1.50	Moderately risky	Moderate alarm
5	1.51–2.00	Risky	High alarm
6	>2	Very risky	Severe alarm

**Table 3 ijerph-16-04805-t003:** Changes of ecological footprint in study area from 2006 to 2016 (ha/cap).

Year	Cropland	Forestland	Grassland	Water	Fossil Energy Land	Build-Up Land	*ef*	EF (ha)
2006	1.33	0.64	1.80	0.07	1.79	0.07	5.71	1.51 × 10^7^
2007	1.27	0.69	1.75	0.07	1.85	0.06	5.69	1.55 × 10^7^
2008	1.63	0.64	2.00	0.07	2.50	0.06	6.90	1.88 × 10^7^
2009	1.87	0.67	2.12	0.08	2.63	0.04	7.39	2.01 × 10^7^
2010	2.24	0.55	2.24	0.08	2.94	0.17	8.22	2.23 × 10^7^
2011	2.27	0.39	2.27	0.09	3.99	0.26	9.26	2.50 × 10^7^
2012	2.65	0.42	2.41	0.09	4.23	0.13	9.93	2.52 × 10^7^
2013	2.72	0.37	2.23	0.09	4.64	0.15	10.21	2.75 × 10^7^
2014	2.99	0.39	2.51	0.08	5.29	0.18	11.43	2.89 × 10^7^
2015	3.10	0.30	2.33	0.10	5.14	0.25	11.23	2.84 × 10^7^
2016	3.09	0.19	2.45	0.10	4.95	0.26	11.04	2.79 × 10^7^

**Table 4 ijerph-16-04805-t004:** Changes of ecological capacity in study area from 2006 to 2016 (ha/cap).

Year	Cropland	Forestland	Grassland	Water	Build-Up Land	Biodiversity Conservation Area	*ec*	Total EC (ha)
2006	2.15	5.04	0.30	0.03	0.22	0.93	6.81	1.84 × 10^7^
2007	2.11	4.98	0.35	0.04	0.25	0.93	6.79	1.85 × 10^7^
2008	2.50	4.97	0.29	0.04	0.27	0.97	7.10	1.94 × 10^7^
2009	2.84	5.09	0.28	0.04	0.41	1.04	7.62	2.07 × 10^7^
2010	3.17	4.95	0.28	0.04	0.47	1.07	7.84	2.13 × 10^7^
2011	2.88	4.97	0.28	0.04	0.43	1.03	7.56	2.05 × 10^7^
2012	3.37	5.30	0.30	0.04	0.49	1.14	8.36	2.12 × 10^7^
2013	3.15	4.98	0.28	0.04	0.46	1.07	7.84	2.11 × 10^7^
2014	3.41	5.31	0.30	0.04	0.50	1.15	8.41	2.13 × 10^7^
2015	3.58	5.32	0.30	0.04	0.53	1.17	8.59	2.17 × 10^7^
2016	3.83	5.31	0.30	0.04	0.57	1.21	8.85	2.24 × 10^7^

**Table 5 ijerph-16-04805-t005:** Changes of ecological security in Hulunbeir grassland from 2006 to 2016.

Year	Ecological Pressure Index	Ecological Security Grade	Characterization State	Ecological Security Alarm Level
2006	0.83	3	Moderately safe	Low alarm
2007	0.84		Moderately safe	
2008	0.97		Moderately safe	
2009	0.97		Moderately safe	
2010	1.05	4	Moderately risky	Moderate alarm
2011	1.22		Moderately risky	
2012	1.19		Moderately risky	
2013	1.30		Moderately risky	
2014	1.36		Moderately risky	
2015	1.31		Moderately risky	
2016	1.25		Moderately risky	

**Table 6 ijerph-16-04805-t006:** Correlation coefficient matrix between independent variables.

Index	A	T	P	U	C
A- per capita GDP	1	0.905 **	−0.811 **	0.816 **	−988 **
Significance test		0.000	0.002	0.002	0.000
T-proportion of the second industry	0.905 **	1	−587	0.561	−0.882 **
Significance test	0.000		0.057	0.072	0.000
P- year-end resident population	−0.811 **	0.587	1	−0.955 **	0.800 **
Significance test	0.002	0.057		0.000	0.003
U- urbanization rate	0.816 **	0.561	−0.955 **	1	−0.814 **
Significance test	0.002	0.072	0.000		0.002
C- unit GDP energy consumption	−0.988 **	0.882 **	0.800 **	−0.814 **	1
Significance test	0.000	0.000	0.003	0.002	

** Correlation is significant at the 0.01 level (2-tailed).

**Table 7 ijerph-16-04805-t007:** Extraction results of principle component characteristic value and contribution rate.

Component	Initial Eigenvalues			Extraction Sums of Squared Loadings			Rotate Sums of Squared Loadings		
	Total	% of Variance	Cumulative%	Total	% of Variance	Cumulative%	Total	% of Variance	Cumulative%
*F* _1_	4.199	83.979	83.979	4.199	83.979	83.979	2.445	48.908	48.908
*F* _2_	0.668	13.354	97.332	0.668	13.354	97.332	2.421	48.425	97.332

**Table 8 ijerph-16-04805-t008:** Rotational component matrix and principal component score coefficient matrix.

Category	Rotational Component Matrix		Principal Component Score Coefficient Matrix	
Indicator	Component		Component	
	*F* _1_	*F* _2_	*F* _1_	*F* _2_
*LnA*	0.481	0.874	−0.135	0.459
*LnT*	0.253	0.957	−0.384	0.675
*LnP*	−0.923	−0.339	0.584	−0.286
*LnU*	0.929	0.344	0.585	−0.284
*LnC*	−0.660	−0.713	−0.121	−0.206

**Table 9 ijerph-16-04805-t009:** Analysis coefficient of principal component regression.

Component	Unstandardized Coefficients	Sig
B	Std. Error
*F* _1_	0.795	0.477	0.134
*F* _2_	1.160	0.091	0.000
R^2^	0.976		
Adjusted R^2^	0.970		
F-statistic	160.668		
Sig.	0.000

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
