# Peer review of "Ecological Security Assessment Based on Ecological Footprint Approach in Hulunbeir Grassland, China"

_ijerph, 2019, doi:10.3390/ijerph16234805_

Round 1
Reviewer 1 Report
The author can shorten the introduction by highlighting the main points only and give emphasis on why this particular study is important.The methods section also described elaborately.It would be a good idea if the author can manage to give a description of the research area in one paragraph. Furthermore, the authors can state the limitation of this study in the discussion section and further research in this area.All in all, the manuscript quality will be improved,if it can be edited by a native English speaker.
Good luck!
Reviewer 2 Report
Dear editor and authors
The manuscript entitled “Ecological security assessment based on ecological footprint approach in Hulunbeir grassland, China (IJERPH-613144)”, utilized the ecological footprint (EF) framework to calculate ecological indicators, EPI, ECC, and EFDI, to give a comprehensive ecological security evaluation. The STIRPAT model was further used to find out the driving factors of ecological deficit under socio-economic development. My major comments are listed below.
1. The manuscript was not well structured neither organized, besides there is no scientific novelty for the temporal calculations of indicators and the final suggestions for reducing human activities.
(1) L343-350: For the PCA paragraph, it is odd to include additional analysis in the section of “Results and Discussion” without any introduction in Methods.
(2) L222: The STIRPAT modelling approach
How do you evaluate the model performance and parameters?
(3) The writing will still need lots of edit.
L16: “… per capita EC … ”
Please check if the terms of ECC and EC are consistent.
L46: “… to the research of internal relationship …”
It should be revised “… to the interactions between them, ”
L51-55: “Among the quantitative methods for … … in the study area.”
The sentence is too long to understand, please rephrase it.
L80-81: “… the researchers concerning ecological security evaluation in … ”
I suggested the sentence revised as “… the evaluation of ecological security in … ”
L108: “about 1000 plant resources”
1000 plant species?
L279: “Furthermore, the cropland forest land and build-up land were … ”
It should be “Furthermore, the cropland, forest land and build-up land were…”
L320-321: “Ecological footprint has the … of ecological footprint [44]”.
It is not clear and seems redundant for me.
2. Although the authors used the style “Results and discussion”, there were actually only three citations (44-46) in the section “4.3.” for the entire content. I don’t think it is enough for a general paper to include such a few references.
Please also shorten the Conclusion.
3. The quality of Tables and Figures needs more efforts to be improved.
L138: Table 1.
Please check the caption and the clarify explain the numbers in the brackets of indicators in “Equivalence factor” and “Yield factor”.
Tables 2-9
The captions of tables are extraordinary presented now.
L316-317: The poor quality of Figure 4 needed to be improved.
The three indicators should be divided into three different colors and same as the Y-axis.
L365-366: In Table 7, the detail variables should be listed in the column, and the final F1-F5 are denoted below of each major component.
L368-369: All contents in Table 9 should be re-arranged, for example R2 is 0.976 instead of “LnI = 0.795F1 + 1.160F2 + LnK”
Reviewer 3 Report
Dear author(s),
The topic developed is of interest and relevance. However, there is still some doubt about the overall quality of your paper to be published in IJERPH. It is in no way your methodological approach or your findings and discussion, but alone the yet imperfect style of scientific writing. I think the paper could improve if the following remarks are taken into account:
The objective of the paper should be expressed explicitly both in the Abstract and de Introduction section. I think that the period of the time chosen in your paper is too short to produce conclusive results. Would it be possible to consider a longer period of time? The abbreviation EPI would be better to appear between brackets when it is used in the first time (v.g. in line 193). I think there must be some erratum in the equation 6 (line 220). Where is EFDI? Is there any assumption of proportionality for the estimated coefficients? (line 230). The Discussion section should be improved. This section provides the interpretation of the results in the context of the existing knowledge (i.e., how do the results contribute to what is already known? How far do they break with existing knowledge and prepare new ground?). The results may be discussed by presenting generalizations that arise from them, by explaining extreme or unexpected observations, or by informing the reader about limitations of the methods. The description of the results has already been presented in the Empirical results section. In fact, in this section you do not quote any author. Please, note that the Conclusion section should not repeat what is in the paper or continue the discussion or present new facts.
I think the following reading could help a revision of the text:
McDonald, M. (2018). Climate change and security: Towards ecological security? International Theory, 10(2), 153-180. doi:10.1017/S1752971918000039
Bargaoui, S. A., Liouane, N., & Nouri, F. Z. (2014). Environmental impact determinants: An empirical analysis based on the STIRPAT model. Procedia-Social and Behavioral Sciences, 109, 449-458.
York, R., Rosa, E. A., & Dietz, T. (2003). STIRPAT, IPAT and ImPACT: analytic tools for unpacking the driving forces of environmental impacts. Ecological economics, 46(3), 351-365.
Reviewer 4 Report
Major issues that should be addressed for revising the manuscript 1. Why did the authors use data from 2006 to 2016 for dynamic analysis? 2. How is the land divided in this study? 3. The authors should elaborate on Stochastic Impacts by Regression on Population, Affluence, and Technology (STIRPAT) model in the introduction section. 4. The language of the manuscript should be carefully polished. Minor issues that should be addressed 1. Page 1, line 15: please check “The results showed that: The per capita EF increased by 93.3%”. 2. Page 1, line 16: full name should be used for “EC”. 3. Page 1, line 17: please check “Within this,EF of fossil fuels”. 4. Page 1, line 34: please check “However, as accelerated industrialization and urbanization, growing incomes and population, and consequently consumption for natural resources, ecological security issues are gradually prominent”. 5. Page 1, line 39: please check “This requires explore the relationship between human activities”. 6. Page 2, line 64: please check “Pressure-State-Response(PSR) model [9-11]”. 7. Page 2, line 64: please check “improved[22].Ecological”. 8 .Page 3, line 98: please check “E115°31′-126°04′,”. 9. Page 3, line 99: please check “Figure1”. 10. Page 3, line 102-104: please check “650~1000m”, “-3 ~ 0℃”, “250 ~ 400mm” 11. Page 3, line 113: please check “Baorixile and Yimin in Hulunbeir city were listed in the second batch of 26 national planning coal mining areas in 2006”. 12. Page 3, line 125: please check “1.94km2”. 13. Page 5, line 138: please check “Table 1. and data sources and instruction”. 14. Page 5, line142: please check “Figure2”. 15. Page 6, line 167: “Where EF is the total ecological footprint of the research subjects (ha)”. 16. Page 6, line 172: “Where EC is the total ecological capacity (ha)”. 17. Page 8, line 212: please check “1<ECC≤1.414”. 18. Page 8, line 221: please check “Where is the proportion of land type in regional ef.”. 19. Page 8, line 238: please check “5.71ha/cap in 2006 to 11.04 ha/cap in 2016”. 20. Page 9, line 273: please check “Figure3”. 21. Page 10, line 290: please check “Figure4”. 22. Page 10, line 296: please check “which meant that the degree of security increased from level 3(moderately safe) to level 4(moderately risky)”. 23. Page 14, line 400-401: please check “ research are as follows: The EF of Hulunbeir grassland doubled nearly two times during the”. 24. Please check and correct the tables. 25. Please carefully check and correct the references.Author Response
Please see the attachment.
Round 2
Reviewer 2 Report
L24: "... STIRPAT mode"
model.
Also check it in the manuscript thoroughly (L167).
Author Response
Thanks a lot for the reviewer's reminder.
We have changed the "mode" to "model" in Line 24, and changed the "development mode" to "development patterns" in Line 167.
Reviewer 4 Report
Minor issues that should be addressed for revising the manuscript 1. Page 2, line 60: please check “from1961 to2007”. 2. Page 2 line 90: please check “production capacity [31]”. 3. Page 3, line 111: please check “650~1000m”, “250 ~ 400mm”. 4. Page 6, line 172: please check “Where EF”. 5. Page 6, line 179: please check “Where EC”. 6. Page 8, line 225: please check “Where is”. 7. Page 8, line 235: please check “Where represents”. 8. Page 14 line414:please check “in EF., ”. 9. Page 14, line 452: please check “Conflicts of Interest: The authors declare no conflict of interest.References”. 10. Please check references 11, 15 and 35.Author Response
Thanks a lot for the reviewer's reminder.
Point 1: Page 2, line 60: please check “from1961 to2007”.
Response 1: We have changed the "from1961 to2007" to "from 1961 to 2007" in Line 60.
Point 2: Page 2 line 90: please check “production capacity [31]”
Response 2: We have changed the "production capacity [31]" to "production capacity[31]" in Line 90.
Point 3: Page 3, line 111: please check “650~1000m”, “250 ~ 400mm”
Response 3: We have changed to “650~1000m”, “250 ~ 400mm” to “650-1000 m ”, “350 mm” in Line 111 and Line 113.
Point 4: Page 6, line 172: please check “Where EF”.
Response 4: We have changed to "In the formula above: EF is the" in Line 172.
Point 5: Page 6, line 179: please check “Where EC”.
Response 5: We have changed to "In the formula above: EC is the" in Line 179.
Point 6: Page 8, line 225: please check “Where is”.
Response 6: We have changed to "In the formula above: pi is", in Line 224.
Point 7: Page 8, line 235: please check “Where represents”.
Response 7: We have changed to "In the formula above: I represents", in Line 235.
Point 8: Page 14 line 414:please check “in EF., ”.
Response 8: We have changed to “in EF," in Line 413.
Point 9: Page 14, line 452: please check “Conflicts of Interest: The authors declare no conflict of interest.References”.
Response 9: We have modified the format of the manuscript in Line 451.
Point 10: Please check references 11, 15 and 35.
Response 10: We have modified reference 11 in Line 479, reference 15 in Line 492, and reference 35 in Line 525.